# Development of a Holographic Waveguide with Thermal Compensation for Augmented Reality Devices

**Artem Solomashenko \*, Alexei Kuznetsov** 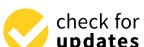**, Vladimir Nikolaev and Olga Afanaseva**

Department of Radioelectronics and Laser Engineering, Bauman Moscow State Technical University,
5/1 2nd Baumanskaya St., 105005 Moscow, Russia
\* Correspondence: asol@bmstu.ru

**Abstract:** In this research, studies were conducted on the possibility of providing thermal compensation of the information display device circuit based on a holographic waveguide when the wavelength of the radiation source ch as a result of changes in ambient temperature. A variant of implementing the waveguide structure in terms of the geometry of the diffraction gratings arrangement is proposed, its main parameters (grating period, thickness, refractive index) and the dependencies between them affecting the quality of the reproduced image are determined.

**Keywords:** holographic waveguide; diffraction grating; wavelength shift

## 1. Introduction

Recently, the technology of holographic waveguides has been gaining particular popularity in the field of information display systems [1–3]. Such waveguides make it possible to present information as virtual images superimposed on objects of the surrounding world in real time, which makes it possible to use them in augmented reality systems, for example, in such areas as geolocation services, navigation systems, visualization systems in medicine, etc. Another promising area in which this technology finds its application are information display devices [4–6]. The principle of operation of such systems is based on the use of a waveguide in which radiation propagates at an angle of total internal reflection (TIR), and then is output using a diffraction optical element (DOE), forming an augmented reality image. Figure 1 shows the simplest case of operation of such systems, when the radiation source is not a microdisplay, but a point LED. It can be noticed that the area of image output to the user (eyebox) increases in comparison with the area of radiation input into the waveguide. Moreover, in this scheme, the holographic waveguide works exclusively with a collimated axial beam, and the observed image is a projection of the radiating area of the radiation source.

The advantage of using a holographic waveguide is the possibility of significantly minimizing the dimensions due to the propagation of radiation inside a thin glass substrate (2–3 mm), as well as reducing the light diameter (up to 5–6 mm) and the focal length of the optical circuit lens while maintaining the size of the exit pupil of at least 20 mm [7,8]. The main components of the information display device, augmented reality and targeting based on a waveguide are: radiation source (LD), collimating system (CS), waveguide (WG). The study of a waveguide in the form of a plane-parallel substrate with three relief-phase diffraction gratings [9] applied to its surface: DOE1 for radiation input relay DOE2 and DOE3 for radiation output. The device, the scheme of which is shown in Figure 1, works as follows: the radiation from LD is collimated using CS, and then falls on DOE1. The radiation diffraction angle at this DOE exceeds the TIR angle, so the radiation is "captured" by the substrate and propagates inside it. With the help of DOE2 and DOE3, the radiation is redirected inside the substrate and output towards the observer. The relay grating DOE2 is oriented at an angle of 45 degrees and is necessary to change the direction of radiation propagation from DOE1 to DOE3 (Figure 1).

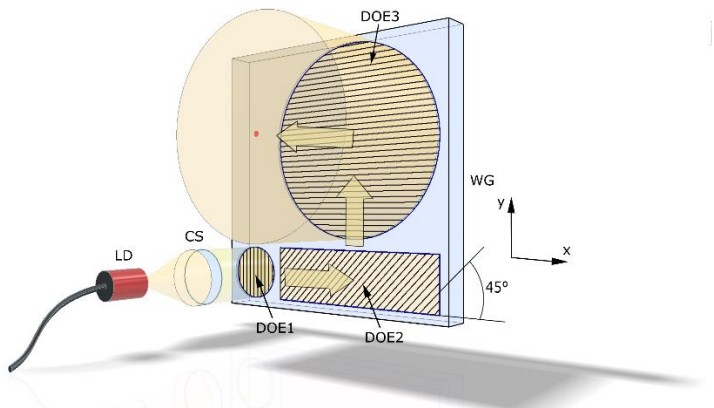

**Figure 1.** The scheme of radiation propagation in a holographic waveguide. DOE1—input area, DOE2—relay area, DOE3—output area.

DOE 2 is necessary to provide two effects—rotation of the radiation by 45 degrees to DOE3, as well as multiplexing (stretching) the pupil area along the horizontal coordinate. When the radiation passes through the substrate along DOE2, each time it hits it, radiation diffraction occurs, as a result of which part of it deviates by 45 degrees and begins to propagate at an angle of total internal reflection towards DOE3, and zero-order diffraction radiation continues to propagate along DOE2 until the next hit on the gratings, etc. In this case, part of the radiation will leave the substrate in the DOE2 region, but it is not possible to avoid these losses. The maximum diffraction efficiency of such a relief-phase DOE does not exceed 34% [10].

This implementation makes it possible to achieve a greater ratio of the areas of the output and input pupils, i.e., the possibility of increasing the output pupil compared to the input one [11]. This is achieved due to the effect of "stretching" the exit pupil: when radiation hits the DOE2 grating, the image expands along the "x" coordinate (Figure 1), and when DOE3 passes, it expands along the "y" coordinate and the subsequent output of radiation towards the observer. In the case of such a scheme, it is possible to achieve pupil expansion with a diameter of 5 mm to a diameter of 20 mm with a size of the relay grating of 5 × 20 mm.

In addition to calculating the parameters of the waveguide itself, the issue of changing the angles of radiation output forming an augmented reality image from a holographic waveguide is of great interest [11,12]. This may be due to inaccuracy of the waveguide itself (wedge shape, profile shape accuracy, etc.), as well as a change in the wavelength of the radiation of the LED or laser diode forming the image. In the future, the article will consider the process of developing the structure of a holographic waveguide that provides thermal compensation.

Thermal compensation in this paper means compensation for changes in the diffraction angles of radiation at the output of the waveguide compared with the angles at the entrance to the waveguide, which is caused by a change in the dominant wavelength of the laser semiconductor module or LED when the ambient temperature changes or the radiation source is heated. Such radiation sources can be part of an augmented reality device with a display requiring backlight, i.e., LCoS, LCD, or DLP type. Data on such sources cited by the manufacturer show that, for example, for LEDs when the ambient temperature changes or when the diode is heated within −20 to 80 degrees, there is a shift of the dominant wavelength by 10–20 nm, and for a semiconductor laser diode with a central wavelength of 635 nm, this shift can be 20 nm Since the operation of the diffraction element, or rather the diffraction angles, are directly related to the wavelength, then for some applications this effect must be taken into account. At the same time, it should be noted that the influence of temperatures on radiation sources in the range of 5 to 35 degrees Celsius (for the environment) was considered, the heating of the LEDs during operation was not

determined. For such temperatures, the instability of the photomaterial and the glass of the waveguide substrate does not manifest itself; therefore, possible fluctuations are caused only by the instability factor of the dominant wavelength of the radiation source.

Such studies were carried out for diffraction gratings recorded in various materials (for example, bichromed gelatin [13], photopolymer [14–16]) and showed the dependence of changes in the parameters and efficiency of the material under the influence of temperature. A rather interesting task is the study of image formation in a holographic waveguide with diffraction gratings based on photosensitive material like photoresist, which have not yet been fully carried out.

## 2. Methods

For correct reproduction of the augmented reality image in the observation plane, it is necessary that the parameters of the waveguide (refractive index, thickness, etc.) be consistent with the DOE structure (period, orientation) [17–19]. One of the main conditions for the correctness of image reproduction is the requirement to overlap the projections of the entrance pupils in the observation plane. When the radiation is introduced, the «0» diffraction order passes through the substrate without being refracted or reflected from the waveguide. In turn, radiation of the +1 diffraction order begins to propagate inside the substrate if the angle of refraction is greater than the angle of total internal reflection. Falling on DOE3, the radiation diffracts and is output to the observer at the same angle at which it was introduced into DOE1 (Figure 2). At the same time, part of the radiation continues to propagate in the waveguide under the action of TIR, and then it is withdrawn from it again, realizing an increase in the size of the exit pupil.

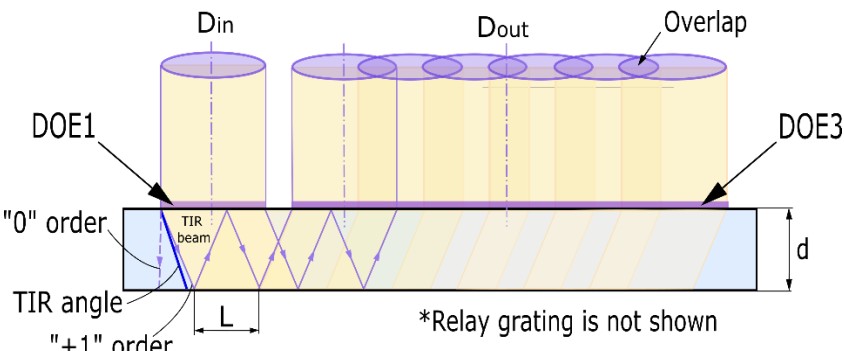

**Figure 2.** Geometry and ray propagation in the substrate: $D_{in}$ is the diameter of the entrance pupil of the substrate, $D_{out}$ is the diameter of the exit pupil of the substrate, L is the beam stoke length.

This implementation makes it possible to minimize the size of the optical system, as noted above, due to the effect of multiplexing the input pupil size Din to the size of the output, for example, Dout = 20 mm.

In order to ensure the propagation of radiation inside the waveguide, it is necessary to select its parameters so that the beam, after hitting the substrate through the input DOE, is refracted at an angle greater than or equal to the angle of total internal reflection [20].

$$\alpha_{TIR} = arcsin\left(\frac{1}{n_g}\right), \tag{1}$$

where $n_g$ represent the refractive index of the substrate glass.

$$L = 2d \cdot tg(\alpha_{TIR}), \tag{2}$$

where d is the thickness of the substrate.

To achieve the overlap of the projections of the entrance pupil in the observation plane, it is necessary that the distance that the beam passes with one reflection is less than or equal to the diameter of the entrance pupil (3).

$$L \leqslant D_{in,} \tag{3}$$

Based on Formulas (1) and (2), it is possible to plot the dependence of the distance that the beam will travel with one reflection on the thickness of the substrate for glasses with different refractive indices (Figure 3), on which the $\alpha_{TIR}$ depends.

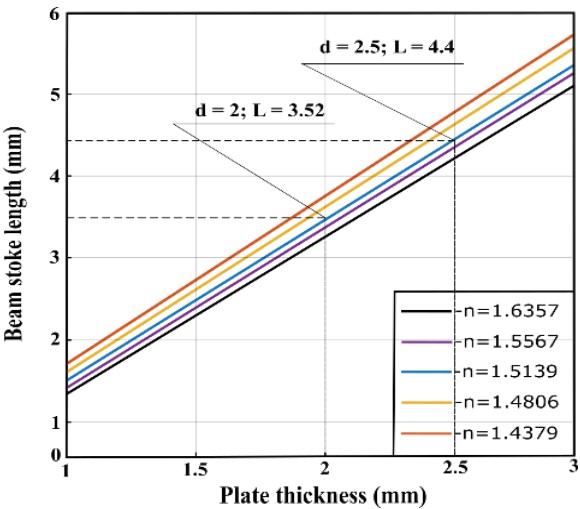

**Figure 3.** The dependence of the beam stoke length on the thickness of the glass for different materials.

In case of non-compliance with condition (3), "blind spots" will appear in the observer's field of view, into which the image is not transmitted, as shown in Figure 4. To study the parameters of the waveguide, a material with the most typical refractive index was selected ($n_g$ = 1.5139). As can be seen from the graph for this material ($\lambda$ = 0.65 μm), the substrate thickness is d = 2 mm, the angle of total internal reflection from (1) is $\alpha_{TIR} = 41°20'$, and the beam stoke length from (2) L = 3.52 mm (Figure 4a).

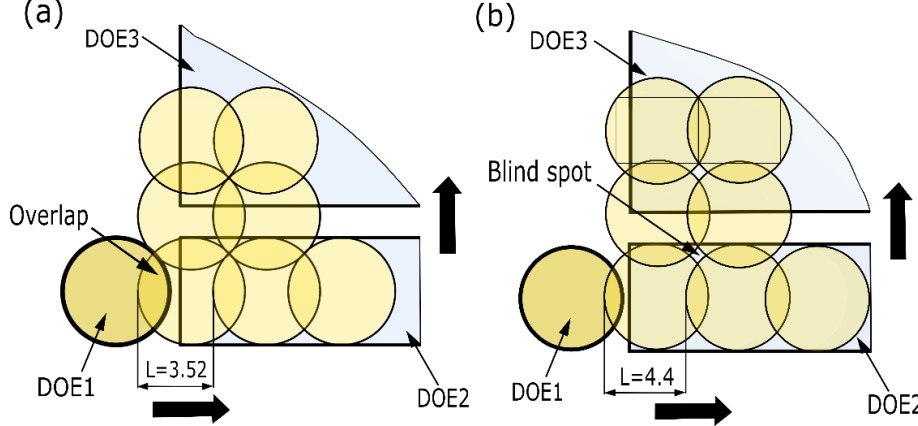

**Figure 4.** Ray tracing in a substrate with a refractive index of the material $n_g$ = 1.5139. (**a**) Substrate thickness d = 2 mm. (**b**) Substrate thickness d = 2.5 mm.

At the same time, if a substrate with a thickness of d = 2.5 mm is used (Figure 4b), "blind spots" will also occur in the observation plane, despite compliance with condition (3).

To eliminate this effect, it is necessary to adjust condition (3) to the form $L \leqslant D_{in}/\sqrt{2}$, i.e., the distance, the distance that the ray passes with one reflection is less than the side of the square with a diagonal equal to $D_{in}$.

In order to calculate the grating period, we use the formula:

$$d_{gr}(n_1 \sin(\alpha_1) + n_g \sin(\alpha_{TIR})) = m\lambda, \tag{4}$$

The paper considers the first order of diffraction ($m = 1$), $n_1 = 1$ (refractive index of air), $\alpha_1 = 0$.

We described relief-phase gratings that form several symmetric diffraction orders at once ("−1" and "+1", "−2" and "+2", etc.). This fact is a feature of this type of diffraction gratings that cannot be eliminated. It should be noted that they have practically no angular and spectral selectivity, so they can work with polychromatic radiation propagating in a wide range of angles, which is important for the formation of an angular field of view in augmented reality devices. Therefore, it is necessary to compromise between the color of the image, the size of the angular field of view and the presence of "parasitic" diffraction orders, which can also form an image or parts of it. At the same time, the diffraction angles of higher orders are quite large (2 times higher than the diffraction angle for "+1" and "−1"); therefore, when diffraction on the introductory grating (DOE 1), the radiation of these orders literally "slides" over the surface of the substrate, almost without undergoing re-reflections, i.e., practically does not leave the substrates of the waveguide to the observer. Another factor is the diffraction efficiency, which is lower in higher orders than in the considered ones, namely: "+1" and "−1"—34%, "+2" and "−2"—10%, "+3" and "−3"—1% [21]. Considering the fact that the radiation diffracts on at least two diffraction gratings during the input and output of radiation (DOE 1 and DOE 3), the total efficiency of the holographic waveguide for such orders does not exceed 1%; therefore, "parasitic" effects from such orders are simply not observed.

The required period of the diffraction grating, which ensures the deviation of the first-order diffraction rays with a wavelength $\lambda = 0.65$ μm by the angle of total internal reflection of the material with a refractive index $n_g = 1.5139$, will be equal to:

$$d_{gr1,3} = \frac{\lambda}{n_g \sin(\alpha_{TIR})} = 0.65 \text{ μm}, \tag{5}$$

We used glass with a refractive index of 1.51 (analogous to BK7 glass). This brand of glass is quite popular, so it was used during research and for recording holographic waveguides. At the same time, it should be noted that the refractive index of glass determines the angle of total internal reflection in the material, i.e., the minimum angle at which radiation can propagate inside the glass substrate without leaving it, so if the task is to increase the size of the angular field of view (i.e., increasing the range of radiation propagation angles in the waveguide), it is recommended to choose glasses with a refractive index of 1.76 and higher.

It is important to consider that the image expands along two coordinates [6]. For the relay grating DOE 2, the expression is valid:

$$d_{gr2} = \frac{d_{gr1,3}}{2cos\beta}, \tag{6}$$

where $\beta$ is the angle of inclination of the relay grating.

It follows from (6) that the period of DOE2 will decrease by a factor of $\sqrt{2}$ due to a change in the trajectory of radiation propagation by 45°, and therefore the period of the relay grating will be equal to $d_{gr2} = 0{,}46$ μm.

Another feature of the development of some optical systems based on a holographic waveguide is the need to consider the range of possible wavelength shift of an LED or laser radiation source within ±10 nm, currently this issue remains unresolved in full [22–24]. The

DOE design methodology should consider such features, since the radiation wavelength of semiconductor laser diodes and the dominant wavelength of the LED changes when its temperature or ambient temperature changes. Therefore, in the previously considered scheme, due to diffraction phenomena, according to (4), an angular displacement of the image on the holographic waveguide occurs, which, accordingly, leads to an inaccuracy in reproducing the line of sight and a shift in the angular position of the output image. This study can be useful for augmented reality systems, in which clear positioning of the output information relative to real-world objects should be provided, for example, in the field of medicine for surgical operations, for displaying contextual prompts during assembly and repair of complex products, for displaying navigation labels and arrows.

As a result of a change in the wavelength of the source by an amount of even 10 nm, the angle of radiation propagation in the waveguide during diffraction at DOE2 (with a period different from DOE1) changes, and therefore the angles of radiation diffraction will change when output from the waveguide. Having solved the system of equations and substituting into it the calculated parameters of the waveguide with a displaced λ = 0.66, we get an angle γ = 1°42′ with a normal to the surface of the substrate (Figure 5a). This is due to the fact that the periods DOE1 and DOE2, in Figure 1, are different, and the angular dispersion (change in the diffraction angle) of the first element cannot be compensated by the similar dispersion of the second.

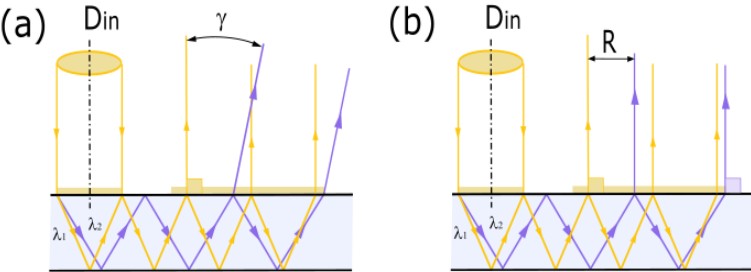

**Figure 5.** (**a**) The angular displacement of the beams in the absence of thermal compensation. (**b**) The displacement of the beams when the wavelength changes by ±20 nm in the scheme with thermal compensation.

Based on the analysis of literary and patent sources, it has been established that this problem can be solved by using hologram compensators, but they have significant weight and size parameters [25]. Another possibility to compensate for the change in the direction of the line of sight and the displacement of the output image when the temperature changes is the use of an alternative structure of the waveguide substrate gratings (Figure 6).

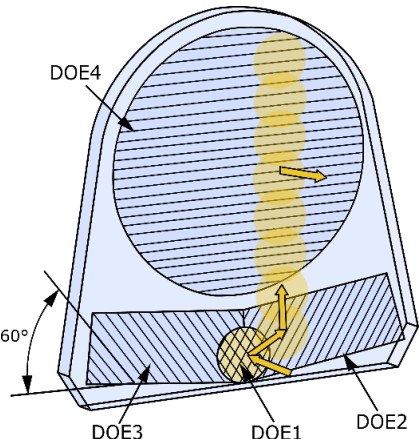

**Figure 6.** Propagation of radiation inside a waveguide with thermal compensation.

In this case, thermal compensation of the holographic waveguide is achieved due to the equality of periods and a change in the orientation of the gratings. DOE1 consists of two diffraction gratings superimposed on each other at angles of 60 and 120 degrees. The left DOE3 and right DOE2 parts of relay DOE are located, respectively, at the conjugate angles of 120 and 60 degrees DOE1. When radiation hits the input area of the DOE1 waveguide, the radiation diffracts on the superimposed structure of DOE1 (superimposed relief-phase gratings) and is divided into 2 parts, one of which spreads along the left side of DOE3, and the other along the right—DOE2. This implementation is widely used in augmented reality devices, because due to the separation of radiation at the input to the waveguide, it allows you to almost double the size of the working angular field of the device compared, for example, with the structure shown in Figure 1.

The angles of inclination of the gratings in DOE 1 and DOE 2 (DOE3) are equal to 60 and 120 (or minus 60) degrees, because in this case, when the radiation is diffracted, a kind of "equilateral triangle" is obtained, in which the step of the radiation inside the substrate does not change when diffracted on the intermediate DOE 2 (DOE3). Due to this fact, the periods of the structure are the same on all DOE. If we consider the situation proposed in Figure 1, then due to the fact that DOE2 (relay grating) is located at an angle of 45 degrees, the step of the sequence changes by a factor of $\sqrt{2}$, therefore, in order to compensate for it, it is necessary to change the period of DOE2. In all cases, except when the angles of the input and relay gratings are 60 and 120 degrees, the periods of the structures will be different. From the point of view of manufacturing the substrate, it is most rational to use the option when all periods are equal, since there is no need to rebuild the stand and the recording scheme.

In the case when the angles of the input and guide gratings coincide, the angular displacement of the beam when hitting DOE1 is compensated by the angular displacement at DOE2 (DOE3) and DOE4, as shown below. DOE in our case are diffraction gratings (DG) [26,27].

Let us write down the radiation diffraction formula [27] for the case of conical diffraction on the introductory grating DOE1 with the angle of inclination of the strokes $\alpha_{GR} = 60$ degrees in general form:

$$\text{k}\left(n_1 sin(\alpha_{inc}) + n_g sin\left(\alpha_{diff}\right)\right) = Kcos(\varphi_{g1}), \tag{7}$$

where $\alpha_{inc}$ is the angle of incidence of radiation on the grating (Figure 7), $\alpha_{diff}$ is the angle of diffraction of radiation, which can be described in spherical coordinates as $sin\left(\theta_{diff}\right)cos\left(\varphi_{diff}\right)$, the wave vector in free space $k = 2\pi/\lambda$, the diffraction grating vector $K = 2\pi/d_{gr}$, $\varphi_{g1}$ is the orientation angle of the diffraction grating DOE1.

$$k\left(sin(\alpha_{inc}) + n_g sin\left(\theta_{diff}\right)cos\left(\varphi_{diff}\right)\right) = Kcos(\varphi_{g1}),$$

$$sin(\alpha_{inc}) = \frac{\lambda}{2d_{gr}} - n_g sin\left(\theta_{diff}\right)cos(\varphi_{diff}), \tag{8}$$

The radiation diffraction equation on the relay DOE2 grating will take the form:

$$k\left(n_g sin\left(\theta_{diff}\right)cos\left(\varphi_{diff}\right) + n_g sin\left(\theta'_{diff}\right)cos\left(\varphi'_{diff}\right)\right) = Kcos(\varphi_{g2}),$$

$$n_g sin\left(\theta_{diff}\right)cos\left(\varphi_{diff}\right) + n_g sin\left(\theta'_{diff}\right)cos\left(\varphi'_{diff}\right) = \frac{\lambda}{2d_{gr}}, \tag{9}$$

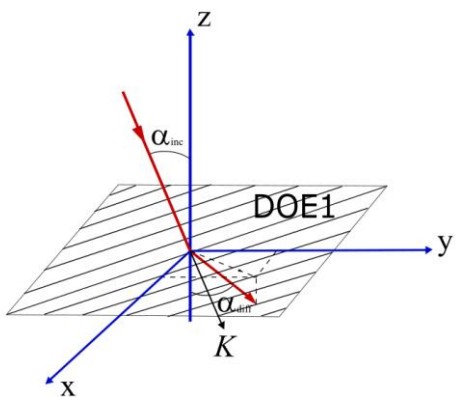

**Figure 7.** The scheme of radiation diffraction on the DOE.

The radiation diffraction equation on the output grating DOE4, i.e., for the angle of the beam of rays $\alpha_{in}$, coming out of the waveguide, will take the form (considering the fact that the angle of the direction of the grating is 180 degrees):

$$k\left(sin(\alpha_{ex}) + n_g sin\left(\theta'_{diff}\right)cos\left(\varphi'_{diff}\right)\right) = Kcos\left(\varphi_{g4}\right),$$

$$sin(\alpha_{ex}) + n_g sin\left(\theta'_{diff}\right)cos\left(\varphi'_{diff}\right) = \frac{\lambda}{d_{gr}}, \tag{10}$$

where $\alpha_{ex}$ is the angle at which the radiation exits the substrate.

Considering the expressions (8) and (9), it is obvious that the output beam is equal to the input:

$$sin(\alpha_{in}) = \frac{\lambda}{2d_{gr}} - n_g sin\left(\theta_{diff}\right)cos\left(\varphi_{diff}\right), \tag{11}$$

Thus, it can be concluded that when the periods of the diffraction gratings on the waveguide are equal, as well as their orientation at angles of 60 and 120 degrees (and in fact, 60 and −60 degrees), the diffraction angle of the radiation introduced into the waveguide is compensated in the event of a change in the wavelength of the radiation source, i.e., the angle of radiation output to the side the pupil of the observer does not change, the radiation exits perpendicular to the surface of the substrate regardless of the wavelength (Figure 5b), and the position of the output image does not shift from the optical axis.

It should be noted that when the wavelength changes, the angle of propagation of the beam inside the waveguide changes, which leads to a slight shift of the light beams at the exit from the substrate (in the observation plane), but this effect does not affect the position of the image.

Based on the proposed structure, we will clarify the main parameters of the holographic waveguide. At the same time, it should be noted that the grating period, determined considering the air defense angle, is determined by the smallest wavelength from the range of its changes, (i.e., $\lambda_{min}$ = 630 nm), and the distance L that the beam passes through is by the largest diffraction angle, i.e., for the largest wavelength ($\lambda_{max}$ = 670 nm). As a result, we obtain a system of equations:

$$\begin{cases} 2d \cdot tg(\alpha_{max}) \leqslant \frac{D_{in}}{\sqrt{2}}, \\ d_{gr} = \frac{\lambda_{min}}{n_g sin(\alpha_{TIR})}. \end{cases} \tag{12}$$

where $\alpha_{max} = arcsin\left(\frac{\lambda_{max}}{nd}\right)$.

As a result, the following parameters were determined for the holographic waveguide: glass with refractive index $n_g$ = 1.5139 ($\lambda$ = 0.65 μm), substrate thickness $d$ = 2 mm, preliminary diameter DOE1 $D_{in}$ = 5 mm, diameter DOE4 $D_{out}$ = 20 mm. Calculated param-

eters: $\alpha_{TIR}$ = 41°20′, L = 3.95 mm, $d_{gr}$ = 0.632 μm. Based on the results obtained, it can be concluded that due to a possible change in the wavelength of radiation, the value of the diameter of the entrance pupil and DOE1 must be increased to a value of 5.6 mm to satisfy condition (12) and the absence of "blind spots" in the observed image.

### 3. Scheme of Obtaining DOE

The DOE in the proposed version are, according to their classification, relief-phase gratings [28] and are obtained in a layer of photosensitive material by analog holography [29,30]. At the same time, the combination of various parameters of diffraction gratings, their number and location on the glass substrate of the holographic waveguide provides the required size of the angular field and the exit pupil, and the number of glass substrates in the holographic waveguide determines the color of the reproduced image.

The technological process of manufacturing holographic waveguides with the above parameters is based on analog recording and consists of the following main stages: control of the glass substrate, application of photosensitive material, exposure (obtaining diffraction gratings), development, finishing operations (cutting and gluing).

The layer of photosensitive material when applied should not exceed 1 μm (from 0.5 nm to 1 μm). The Microposit S1805 positive photoresist manufactured by Shipley is used as a sensitive material (for subsequent recording of gratings). Dilution of the Microposit S1805 photoresist is not carried out, because its viscosity is small enough and allows you to obtain sufficiently thin layers.

The exposure process is a recording of diffraction gratings on a photosensitive material under the influence of laser radiation. To record the gratings, the method of analog holography is used, that is, to obtain an interference pattern of two waves (the angle of incidence of which on a substrate with a photosensitive resistor must be strictly defined) from a single coherent radiation source (in this case, a gas helium-cadmium laser with a power of 120 mW). This method can be implemented by assembling a specialized laser recording stand, the scheme of which is shown below in Figure 8 [31].

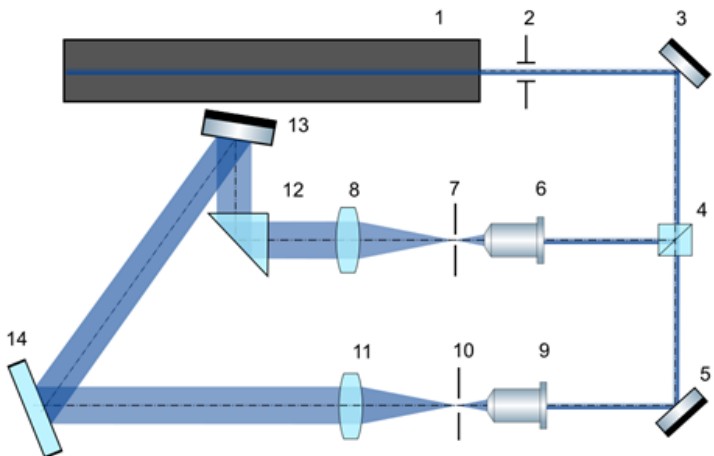

**Figure 8.** The scheme of analog recording of DOE. 1—laser; 2—shutter; 3, 5, 13—mirrors; 4—beam–splitting cube; 6, 9—micro lenses; 7, 10—aperture; 8, 11—lenses, 12—prism (mirror); 14—photographic substrate.

To form a color image, it is necessary to connect three substrates of holographic waveguides, each of which operates mainly at its own wavelength (red, blue or green) [32,33]. Thus, the radiation is divided according to the spectral composition and the angle of incidence at the input by three waveguides, passes through them, obeying the air defense condition, and then, when output from the last DOE, it is summed up and forms an RGB image. The period of each diffraction grating is determined by the recording angle, which is 42, 50 and 57 degrees for the grating periods of 610, 520 and 460 nm, respectively. The

orientation of the grating depends on the angle of rotation of the substrate 14 relative to the two interfering beams during recording, as shown in Figure 8. When recording grids 1a, 2b and 1b, 2a (Figure 6), the substrate should be rotated by an angle of 60 and −60 degrees, respectively, and when recording the output grid, the substrate is oriented along the larger side. The direction of the strokes in each case is perpendicular to the plane of incidence of the subject and reference beam (Figure 9).

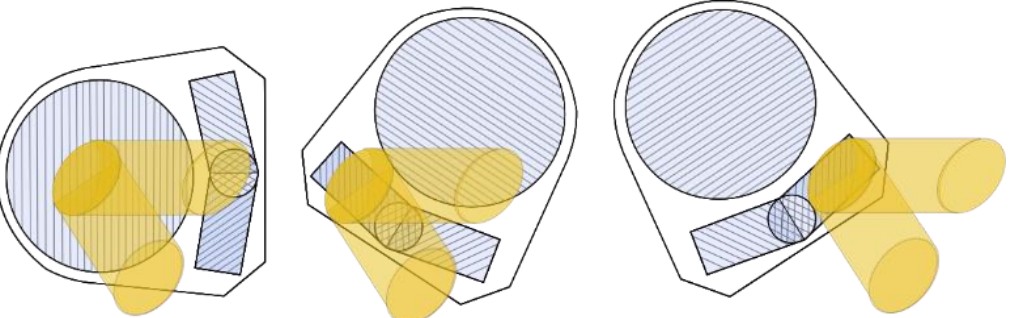

**Figure 9.** The location of the substrate when recording different DOEs.

For the development of the photoresistive layer, the developer Microposit 303 developer is used (Figure 10), diluted to a concentration of 1:7. The low concentration of the developer provides slower etching of the photographic material with the formation of a high-frequency surface relief. Otherwise, the diffraction grating may not appear.

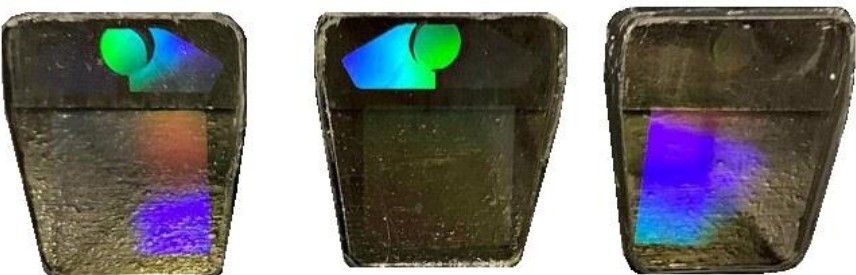

**Figure 10.** Photo of an experimental sample of a holographic waveguide.

The transparency of such an element, consisting of three RGB substrates, is about 65–70%. This transmission is due to the fact that the gratings are obtained in a photoresist layer, which itself has a certain yellowish color, i.e., absorption is present due to the features of the photographic material.

The diffraction efficiency of the obtained individual DOE reaches 24% for R and G substrates and 18% for substrate B (the reduced efficiency is associated with very small grating periods for this wavelength). Thus, the total efficiency of such a waveguide, in which radiation is consistently diffracted on several gratings, is extremely small (several percent). This is due to the fact that the gratings used in holographic waveguides are predominantly thin. Unlike volume gratings with high efficiency, the parameters of which depend on the thickness of the material and the refractive index, and the angular selectivity is very strong and can be several degrees, the effectiveness of a thin grating depends only on the depth of the surface relief. Their angular and spectral selectivity is not pronounced, so they can work with a color image in a wide working angular field (up to 50 degrees). There are promising ways to increase the efficiency of gratings with surface relief, namely: the use of an asymmetric or inclined relief profile or the promising use of a new type of elements—metasurfaces [34–36], but these methods were not considered in this paper.

## 4. Experimental Results and Discussion

To conduct experimental studies of the proposed structure of a holographic waveguide on the issue of providing thermal compensation, the following experimental scheme is proposed.

As a test image, a grid image of blue, green and red on a black background is used, as shown in Figure 11. The image was generated using the optical engine, which is a miniature image projector based on a DPL matrix with LED backlight. The radiation from three RGB LEDs is combined into one branch to obtain white color using dichroic mirrors, and for the best mixing of the three colors, a microlens raster is additionally used.

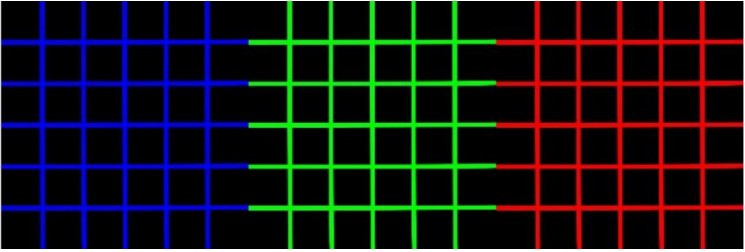

**Figure 11.** Test Images.

The test image is output using a holographic waveguide. The photo of the output image is shown in Figure 12.

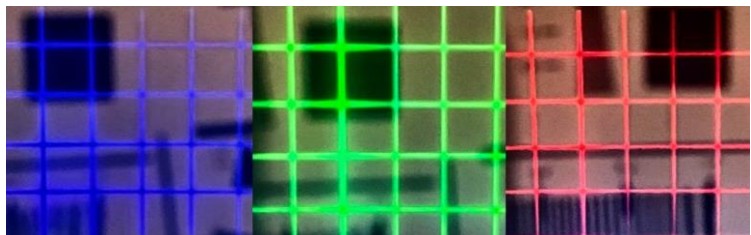

**Figure 12.** An image observed using a holographic waveguide.

To analyze the change in the angles of radiation output from the substrate when the dominant wavelength of the source radiation changes, which accordingly leads to an angular displacement of the output image on a holographic waveguide (all three substrates were stacked), a collimator with a reference grid was used, as shown in Figure 13.

The image output through the waveguide is recorded using a collimator, and its angular displacement is estimated in angular minutes by a reference grid located in the focus of the eyepiece, as shown in Figure 14. In this case, the initial position of a separate line of the test image is fixed on the collimator grid, and then the angular displacement of the position of this line is measured already when the wavelength of the radiation source (LED or laser diode/display) changes.

Figure 14 shows examples of test images, namely: Figure 14a is an image of RGB grids reproduced using a holographic waveguide to evaluate its performance, and Figure 14b,c are images of the grid reproduced "before" and "after" the LED wavelength shift. In the case of a wavelength shift, a small deviation of the test image of the grid relative to the measuring scale is observed, i.e., angular displacement of the image output through the holographic waveguide. Thus, it is shown that the operation of a holographic waveguide, and, accordingly, the position of the displayed image, depends on the change in the wavelength of the radiation source, but can be partially compensated by a certain structure of the arrangement of diffraction gratings on its surface.

The change in the wavelength of the radiation output from the holographic waveguide was recorded by the magnitude of the change in the dominant wavelength of the radiation spectrum using the Jeti Spectraval 1501 spectroradiometer, as shown in Figure 15.

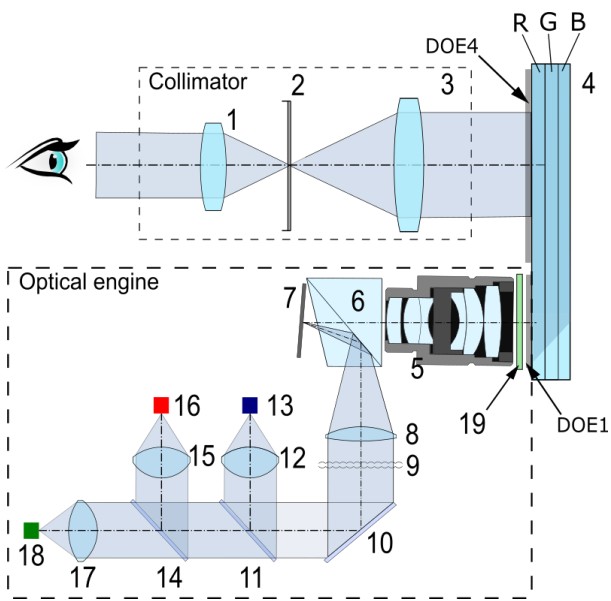

**Figure 13.** The scheme for determining the angular displacement of the output image. 1–3—collimator (2—measuring grid); 4—waveguide; 5–18—optical engine (13, 16, 18—LEDs; 8, 12, 15, 17—lenses; 10, 11, 14–mirrors; 9—microlenses; 6—prism; 7—display; 19—filter only for green LED).

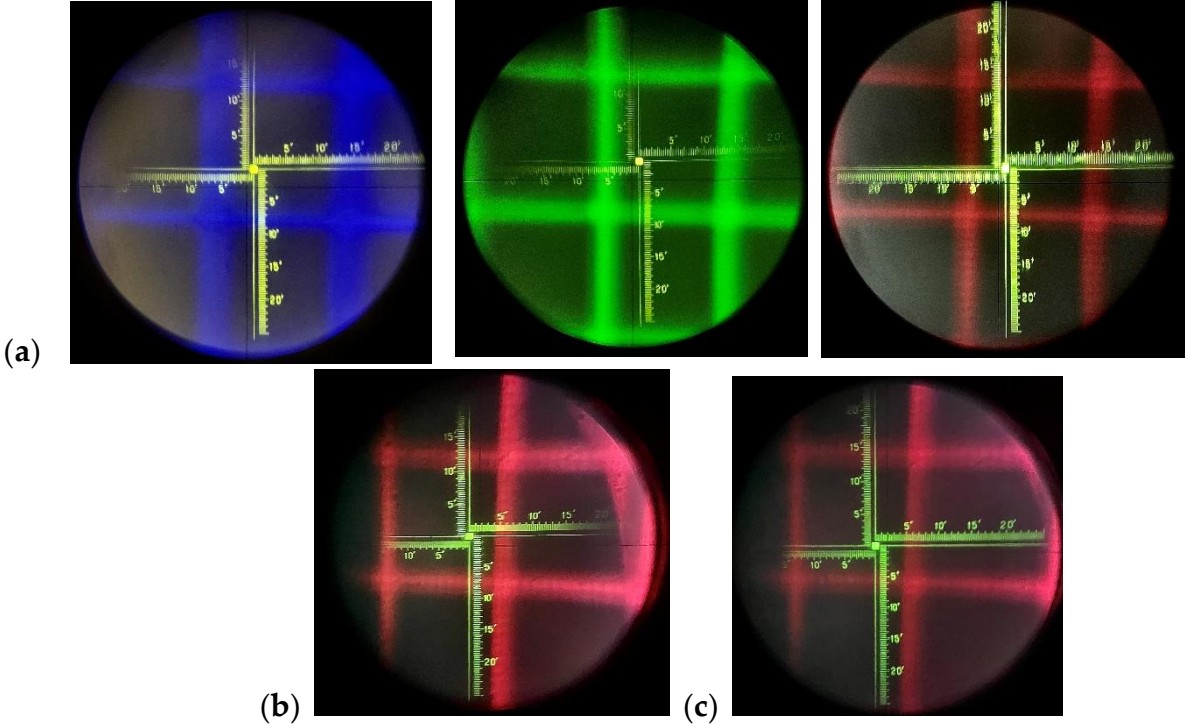

**Figure 14.** (**a**) Test image observed against the background of the collimator grid. (**b**) Wavelength shift is 0.5 angular minutes. (**c**) There is no wavelength shift.

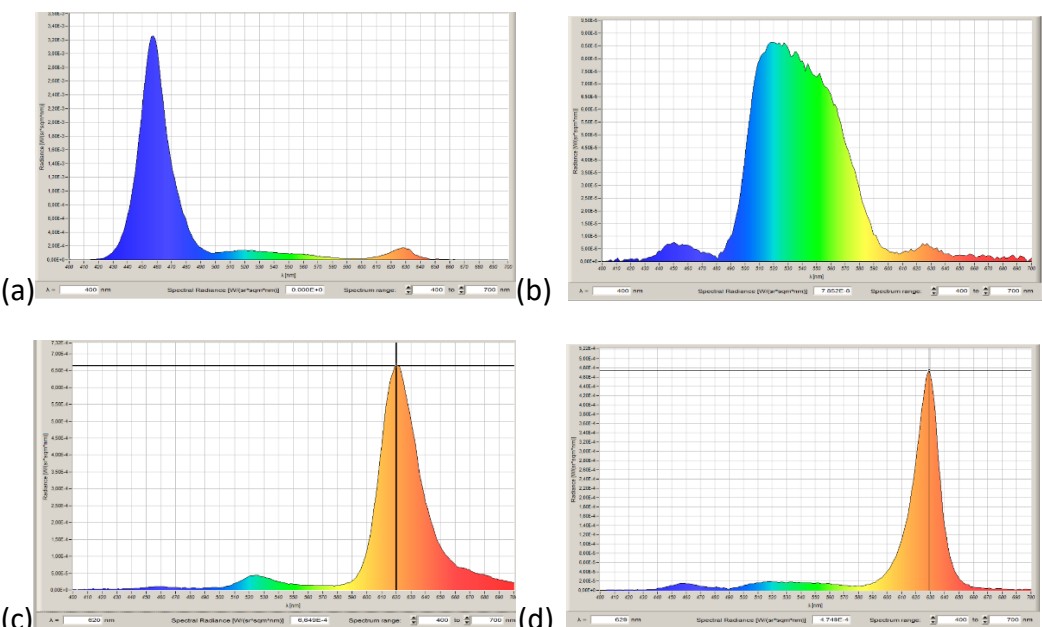

**Figure 15.** Spectra of the output image. (**a**) λ = 457 nm. (**b**) λ = 520 nm. (**c**) λ = 628 nm (at a temperature equal to 20 °C). (**d**) Wavelength offset λ = 618 nm (at a temperature equal to 5 °C).

Figure 15 shows the emission spectra of the RGB LED radiation source used as part of the optical engine. The dominant wavelengths correspond to 457 nm, 520 nm and 628 nm (for the red wavelength, an example of a shift of the dominant wavelength by 10 nm as a result of a temperature change of 15 °C is given). The normal ambient temperature was 20 degrees, and then it was changed by plus / minus 15 degrees Celsius.

At the same time, it should be noted that for the radiation source used, the width of the red and blue spectra is quite small and reaches 12–15 nm, which really allows us to talk about a change in the dominant, i.e., the main wavelength in the spectrum. In the case of a green radiation source, the spectrum width reaches almost 100 nm, so a change in the dominant wavelength by an amount of about 10 nm will not significantly affect the angular displacement of the output image. However, because the green color spectrum initially contains a large set of wavelengths with maximum radiation intensity, then thermal compensation will be provided for the entire wide spectrum of radiation from the green radiation source, i.e., radiation at all wavelengths of green color will come out perpendicular to the surface of the holographic waveguide. In the case of using a radiation source with a narrower radiation spectrum, the thermal compensation condition will also be fulfilled, as in the case of blue and red wavelengths.

As a result, the following dependences of the angular displacement of the output image on the wavelength were obtained, as shown in Figure 16a. For the red and blue wavelengths, a slight angular displacement of the output image is observed—about ±0.5 angular minutes with a shift of the dominant wavelength by ±10 nm. For the spectrum of the green source, several transmission spectral filters were used at the input of radiation into holographic waveguides in order to obtain pronounced maxima within a wide spectrum of the radiation source. In this case, the angular displacement of the test line image was ±1 angular minute. This change is extremely small, so we can talk about the effect of thermal compensation of this scheme. In Figure 16b shows the dependence of the angular displacement on the change in wavelength for the scheme indicated in Figure 1. As can be seen from the graphs, the angular displacement of the image in the output region is more than one degree for all wavelengths, which indicates the absence of thermal compensation.

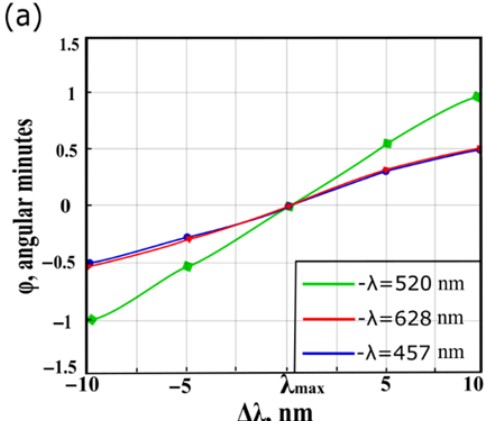
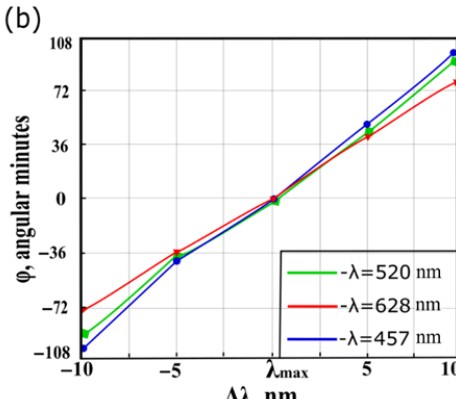

**Figure 16.** A graph of the dependence of the angular displacement of the output image on the change in the wavelength of radiation. (**a**) A circuit with thermal compensation. (**b**) A circuit without thermal compensation.

In practice, an angular displacement of 1 degree at an image output distance of 0.6 m from the user corresponds to a linear displacement of 10 mm, and for a distance of 1.5 m–about 25 mm. Of course, this angular displacement is not very large, but, as we noted, it may be significant for some applications. For example, for the field of medicine, when in augmented reality mode, three-dimensional models of his organs, arteries or veins can be superimposed on the patient's body during surgical operations to remove tumors or when administering medications (injections). The signature really did not sound quite right and was corrected.

As can be seen from Figure 16, the above scheme for implementing a holographic waveguide based on relief-phase diffraction gratings (DOEs) of the same period, two of which are located at an angle of 60 degrees relative to each other, provides almost complete compensation for the angular displacement of the output image when the wavelength of the radiation source changes, which can occur, for example, when the temperature changes environment.

It is important to note that a slight angular displacement of the output image was observed only in the vertical plane, i.e., in the plane perpendicular to the strokes of the output grid DOE4, while no image displacement was detected along the horizontal axis. Indeed, angular displacement in the vertical plane, i.e., in the plane perpendicular to the grating strokes on which diffraction occurs, is present. Most likely, this is due to inaccuracies in the manufacture of DOE (angular mismatch of DOE1 and DOE2) on the recording stand, but the permissible value of these inaccuracies was not evaluated in this study, because the obtained values of angular displacement are quite small. In the horizontal plane, i.e., in the plane of the grating strokes, this effect is not observed.

Resolution, the size of field of view, brightness and unevenness of the image brightness within the eyebox, there is no noticeable deterioration in image quality (it only shifts). This is due to the fact that the diffraction gratings that we manufacture are relief-phase [30], i.e., they do not have spectral selectivity and will work equally with all incident beams of rays (deflect them, remove them from the substrate, etc.), regardless of the displacement of the dominant wavelength.

Thus, for augmented reality systems based on holographic waveguides of this type, regardless of external conditions, the image will be displayed correctly.

## 5. Conclusions

The research analyzes, selects and calculates the main parameters of a holographic waveguide, taking into account the issue of thermal compensation when the wavelength of the radiation source changes in order to preserve the position of the optical axis and minimize the possible angular displacement of the augmented reality image.

The waveguide structure is proposed in the form of four DOEs with the same periods: the input DOE1 and the relay DOE2 and DOE3 superimposed on it have angles of inclination of 60° and 120°, the output DOE4 is located at an angle of 180°, which makes it possible to compensate for the displacement of λ within ±20 nm.

This study is useful for augmented reality systems, in which a clear positioning of models and output information about real-world objects should be provided, for example, in the field of medicine for surgical operations, for displaying contextual prompts during assembly and repair of complex products, for displaying navigation labels and arrows.

**Author Contributions:** Conceptualization, A.S. and O.A.; methodology, A.S., O.A. and V.N.; verification, A.S., O.A., V.N. and A.K.; formal analysis, A.S.; research, A.S. and O.A.; resources, V.N. and A.K.; writing—preparation of the original project, A.S., O.A. and V.N.; writing—review and editing, A.S. and O.A.; visualization, A.S.; supervision, V.N.; project administration, A.K. and A.S.; fundraising, A.K. All authors have read and agreed to the published version of the manuscript.

**Funding:** This research was funded by the Priority 2030 program at the Bauman Moscow State Technical University.

**Institutional Review Board Statement:** Not applicable.

**Informed Consent Statement:** Not applicable.

**Data Availability Statement:** Data are available from the authors upon request.

**Acknowledgments:** Not applicable.

**Conflicts of Interest:** The authors declare no conflict of interest.

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
