# Peer review of "Development of a Holographic Waveguide with Thermal Compensation for Augmented Reality Devices"

_applsci, doi:10.3390/app122111281_

Round 1

Reviewer 1 Report

The paper studies the design requirements of diffractive gratings for correct reproduction of augmented reality image. System equations for achieving absence of blind spots are derived, and multiple surface-relieve gratings are designed to minimize wavelength shift. The authors further verified the study by experimentally demonstrating a holographic waveguide with small angular displacement.

Overall, the studies on image formation in relation to blind spot and chromatic aberration are useful in AR engineering. The experiments are also in good agreement with the theoretical prediction. However, I have some concerns that need to be addressed prior to any decision:

 1.   The claim of  “thermal compensation” is confusing, since no thermal fluctuations have been considered either in the design or experiment. The studies appear to be more focused on image formation and aberration analyses. While the photoresist may experience thermal instability, but only around or above the temperature of 100 deg, which doesn't apply in the ambient environment. Thus, the authors should better clarify the term “thermal compensation”.

2.  The paper falls short of explaining the device efficiency. What is the diffraction efficiency of the grating? How does the refractive index, duty cycle and thickness affect the grating efficiency?

3.   Another important metric of AR displays is the see-through property. Therefore, the average transmission of the gratings should been provided.

4.    The paper demonstrates DOE based on surface-relief gratings. It’s worth noting that a new type of diffractive optics called metasurfaces has emerged as an attractive platform for AR and holography, which might be interesting to a broader research community. These include:

 a. Li, Zhaoyi, et al. "Meta-optics achieves RGB-achromatic focusing for virtual reality." Science Advances 7.5 (2021): eabe4458.

b. Lee, Gun-Yeal, et al. "Metasurface eyepiece for augmented reality." Nature communications 9.1 (2018): 1-10.

c.   Zhou, You, et al. "Multifunctional metaoptics based on bilayer metasurfaces." Light: Science & Applications 8.1 (2019): 1-9.

Reviewer 2 Report

The authors presented an investigation of a themal compensation for holographic waveguides for usage in argrumented reality applications.

There are a few points which should be described more detailed:

- The authors investigate the influence of the surrounding temperature to an LED-based imaging setup. The authors did not mention the absolute temperature or the temperature of the LED itself. Does a change of +- 15°C really infect the LED which can have working temperatures of ~80° (or even more, depending on the type). More information are required.

-Regarding DOE2 (figure 1): It is not clear how the reflection works in detail. Is the DOE coated or are the outcoupling losses neglected? Sketch and information about the efficiency of the DOEs would be very helpful.

- line 104: The authors mention that only first order diffractions were considered. What happens with higher order diffractions in the "real life". Do they lead to a blurred image or do they have no effect on the image due to leaking out of the substrate?

-line 146 (figure 6): For an easier understanding should the parts 1a, 1b... also described in the caption.

- line 176: What does alpha_ex mean? Description missing.

- line 198: The authors calculated exactly which refractive index they are needing for the substrate. The information which glass they used is missing.

- line 232: Does the authors also stacked the holograms or investigated them separately?

line 258: How were the image achieved? SLM, amplitude mask...?

line 283: What kind of source is used? Different chips? One white light LED where the single chips can be addressed or different filters on the LED?

line 305: Why does the authors used a filter here? Is this also necessary for the other wavelengths? Sketch of the experimental setup would be helpful.

line 314: Is an angular displacement of +-1° critical for augmented reality applications? Which wavelength shift correlates with which temperature? "Circuit with thermal compensation" sound like direct cooling of the LED but I think the author meant the DOE-setup.

line 324: Why is there only an angular displacement in the vertical plane? Does the DOE2 and 3 are not fabricated as recommended? Other influences?

Reviewer 3 Report

Please see the attached pdf 

Reviewer 4 Report

This article studied the impact on waveguide display’s image angular position shift of wavelength change when light sources’ temperature change, and proposed a thermal compensation method based on designed waveguide’s geometry structure. I believe that this research is meaningful in many augmented display applications, for example, as the authors suggested, in surgical operations, industrial assembly and vehicle navigation. So, it is a theme that worth studying and publishing. However, for the work proposed in the article, there are some questions about the method and results which I think are not discussed sufficiently enough and need to be explained more by the authors. So, I recommend that this article reviewed after a major revision.

The questions are explained in detail below.

1. How is the value of angular shift γ obtained in Fig. 5(a)? Furthermore, how the angular shift is analyzed for a specific waveguide structure? Please describe the numerical method in more detail in the article.

2. In the waveguide structure shown in Fig. 6, the DOE 1 consists of two superimposed gratings with angles of 60 and 120 degrees, and the DOE 2 consists of two parts which located with angles of 120 and 60 degrees. How these values of angles decided? Or why choose them? Are there any other possible alternative values can be chosen to carry out thermal compensation? If yes, how to make a best choice? Please add related explanations.

3. Is the display images of the waveguide system shown in Fig. 14 a result after thermal compensation or not? Please indicate that clearly in the figure, and add comparison of display images results before and after thermal compensation.

4. The recorded light wave in Fig. 8 are not guided in a total internal reflection (TIR) way. How to guarantee that the diffracted light is transform in a TIR way when the fabricated waveguide is used? And will this recording method cause unwanted diffraction orders, or how to avoid this problem?

5. Dose angular shift of light rays under wavelength shift cause displayed image’s quality decreased? Could the thermal compensation method solve this problem as well as image position shift problem?

Round 2

Reviewer 2 Report

I thank the authors for the detailed response to all the comments. I recommend the submitted manuscript for publication in the journal.

Author Response

Description in the text (lines 388-396) about additional pictures in fig. 14(b) and (c) have been added to make the article more readable.

Reviewer 4 Report

The authors have provided sufficient answers for the questions. The revised manuscript is good and deserves publication on Applied Sciences after a minor correction described below. I suggest that this manuscript is accepted after the minor correction is done.

Line 385, the word “wavelength” is spelt mistakenly as “wavelenght”.

And I also suggest the authors do a little more description in text about the added pictures in Fig. 14 (b) and (c), so that the article will be more readable.

Author Response

We sincerely thank the Reviewer for the time and assessment of the work. The manuscript has been revised according to your comments.

Point 1: Line 385, the word “wavelength” is spelt mistakenly as “wavelenght”.

Response 1: Sorry, that's a typo. Thank you for paying attention to this.

Point 2: And I also suggest the authors do a little more description in text about the added pictures in Fig. 14 (b) and (c), so that the article will be more readable.

Response 2: The following description of the figures has been added to the manuscript:

Figure 14 shows examples of test images, namely: fig. 14a is an image of RGB grids reproduced using a holographic waveguide to evaluate its performance, and fig. 14b and 14c are images of the grid reproduced "before" and "after" the LED wavelength shift. In the case of a wavelength shift, a small deviation of the test image of the grid relative to the measuring scale is observed, i.e. angular displacement of the image output through the holographic waveguide. Thus, it is shown that the operation of a holographic waveguide, and, accordingly, the position of the displayed image, depends on the change in the wavelength of the radiation source, but can be partially compensated by a certain structure of the arrangement of diffraction gratings on its surface.

Changes are presented in manuscript on lines 388-396.

Kind regards, Authors.